# Incorporating BERT into
# Parallel Sequence Decoding with Adapters

**Junliang Guo[1], Zhirui Zhang[2], Linli Xu[1,3]\*, Hao-Ran Wei[2], Boxing Chen[2], Enhong Chen[1]**
[1]Anhui Province Key Laboratory of Big Data Analysis and Application,
School of Computer Science and Technology, University of Science and Technology of China
[2]Alibaba DAMO Academy
[3]IFLYTEK Co., Ltd.
[1]`guojunll@mail.ustc.edu.cn, {linlixu, cheneh}@ustc.edu.cn`
[2]`{zhirui.zzr, funan.whr, boxing.cbx}@alibaba-inc.com`

## Abstract

While large scale pre-trained language models such as BERT [5] have achieved great success on various natural language understanding tasks, how to efficiently and effectively incorporate them into sequence-to-sequence models and the corresponding text generation tasks remains a non-trivial problem. In this paper, we propose to address this problem by taking two different BERT models as the encoder and decoder respectively, and fine-tuning them by introducing simple and lightweight *adapter* modules, which are inserted between BERT layers and tuned on the task-specific dataset. In this way, we obtain a flexible and efficient model which is able to jointly leverage the information contained in the source-side and target-side BERT models, while bypassing the catastrophic forgetting problem. Each component in the framework can be considered as a plug-in unit, making the framework flexible and task agnostic. Our framework is based on a parallel sequence decoding algorithm named Mask-Predict [8] considering the bi-directional and conditional independent nature of BERT, and can be adapted to traditional autoregressive decoding easily. We conduct extensive experiments on neural machine translation tasks where the proposed method consistently outperforms autoregressive baselines while reducing the inference latency by half, and achieves 36.49/33.57 BLEU scores on IWSLT14 German-English/WMT14 German-English translation. When adapted to autoregressive decoding, the proposed method achieves 30.60/43.56 BLEU scores on WMT14 English-German/English-French translation, on par with the state-of-the-art baseline models.

## 1 Introduction

Pre-trained language models [26, 27, 5, 39] have received extensive attention in natural language processing communities in recent years. Generally, training of these models consists of two stages. Firstly, the model is trained on a large scale monolingual corpus in a self-supervised manner, and then fine-tuned end-to-end on downstream tasks with task-specific loss functions and datasets. In this way, pre-trained language models have achieved great success on various natural language understanding tasks such as reading comprehension and text classification, and BERT [5] is one of the most successful models among them.

While fine-tuning BERT for common language understanding tasks is straightforward, for natural language generation which is one of the core problems in NLP [2, 34, 24], how to incorporate BERT

---

remains substantially challenging. We conclude the main challenges as three-fold considering that the sequence-to-sequence framework [33] is the backbone model of generation tasks. On the encoder side, as studied in [41], simply initializing the encoder with a pre-trained BERT will actually hurt the performance. One possible explanation could be that training on a complex task with rich resources (e.g., machine translation) leads to the catastrophic forgetting problem [23] of the pre-trained model. On the decoder side, which can be treated as a conditional language model, it is naturally non-trivial to marry unconditional pre-training with conditional fine-tuning. And the bidirectional nature of BERT also prevents it from being directly applied to common autoregressive text generation. In addition, fine-tuning the full model is parameter inefficient considering the enormous scale of recent pre-trained language models [28] while being unstable and fragile on small datasets [20].

To tackle these challenges, in this paper, we propose a new paradigm of incorporating BERT into text generation tasks under the sequence-to-sequence framework. Specifically, we construct our framework based on the following steps. We first choose two pre-trained BERT models from the source/target side respectively, and consider them as the encoder/decoder. For example, on the WMT14 English-German machine translation task, we take `bert-base-cased` as the encoder and `bert-base-german-cased` as the decoder. Then, we introduce lightweight neural network components named *adapter* layers and insert them into each BERT layer to achieve the adaptation to new tasks. While fine-tuning on task specific datasets, we freeze the parameters of BERT layers and only tune the adapter layers. We design different architectures for adapters. Specifically, we stack two feed-forward networks as the encoder adapter, mainly inspired from [3]; and an encoder-decoder attention module is considered as the decoder adapter. Considering that BERT utilizes bi-directional context information and ignores conditional dependency between tokens, we build our framework on a parallel sequence decoding algorithm named Mask-Predict [8] to make the most of BERT and keep the consistency between training and inference.

In this way, the proposed framework achieves the following benefits. 1) By introducing the adapter modules, we decouple the parameters of the pre-trained language model and task-specific adapters, therefore bypassing the catastrophic forgetting problem. And the conditional information can be learned through the cross-attention based adapter on the decoder side; 2) Our model is parameter efficient and robust while tuning as a benefit from the lightweight nature of adapter modules. In addition, thanks to parallel decoding, the proposed framework achieves better performance than autoregressive baselines while doubling the decoding speed; 3) Each component in the framework can be considered as a plug-in unit, making the framework very flexible and task agnostic. For example, our framework can be adapted to autoregressive decoding straightforwardly by only incorporating the source-side BERT encoder and adapters while keeping the original Transformer decoder.

We evaluate our framework on various neural machine translation tasks, and the proposed framework achieves 36.49/33.57 BLEU scores on the IWSLT14 German-English/WMT14 German-English translation tasks, achieving 3.5/0.88 improvements over traditional autoregressive baselines with half of the inference latency. When adapting to autoregressive decoding, we achieve 30.60/43.56 BLEU scores on the WMT14 English-German/English-French translation tasks, on par with the state-of-the-art baseline models.

## 2   Background

**Pre-Trained Language Models**   Pre-trained language models aim at learning powerful and contextual language representations from a large text corpus by self-supervised learning [26, 5, 27, 28, 39, 6, 22], and they have remarkably boosted the performance of standard natural language understanding tasks such as the GLUE benchmark [35]. BERT [5] is one of the most popular pre-training approaches, whose pre-training objective consists of masked language modeling (MLM) and next sentence prediction. The idea of MLM has been applied widely to other tasks such as neural machine translation [8]. Given an input sentence $x = (x_1, x_2, ..., x_n)$, MLM first randomly chooses a fraction (usually $15\%$) of tokens in $x$ and substitutes them by a special symbol `[MASK]`, then predicts the masked tokens by the residual ones. Denote $x^m$ as the masked tokens and $x^r$ as the residual tokens, the objective function of MLM can be written as:

$$L_{\text{MLM}}(x^m|x^r; \theta_{\text{enc}}) = -\sum_{t=1}^{|x^m|} \log P(x_t^m|x^r; \theta_{\text{enc}}),  \tag{1}$$

where $|x^m|$ indicates the number of masked tokens.

Among the alternative pre-training methods, UniLM [6] extends BERT with unidirectional and sequence-to-sequence predicting objectives, making it possible to fine-tune the pre-trained model for text generation tasks. XLM [19] achieves cross-lingual pre-training on supervised parallel datasets with a similar objective function as MLM. MASS [30] proposes a sequence-to-sequence monolingual pre-training framework where the encoder takes the residual tokens $x^r$ as input and the decoder predicts the masked tokens $x^m$ autoregressively. BART [21] adopts a similar framework and trains the model as a denoising autoencoder. Although achieving impressive results on various text generation tasks, these models are equipped with large-scale training corpora, therefore are time and resource consuming to train from scratch. In this paper, we focus on leveraging public pre-trained BERT models to deal with text generation tasks.

**Incorporating Pre-Trained Models**    There exist several recent works trying to incorporate BERT into text generation, which are mainly focused on leveraging the feature representation of BERT. Knowledge distillation [15, 18] is applied in [37, 38, 4] to transfer the knowledge from BERT to either the encoder [38] or decoder side [37, 4]. Zhu et al. [41] introduces extra attention based modules to fuse the BERT representation with the encoder representation. Most of these methods only incorporate BERT on either the source side or the target side. Our framework, on the other hand, is able to utilize the information of BERT from both sides.

**Fine-Tuning with Adapters**    Adapters are usually light-weight neural networks added into internal layers of pre-trained models to achieve the adaptation to downstream tasks, and have been successfully applied to fine-tune vision models [29], language models [16, 36] and multilingual machine translation models [3, 17]. Different from these works, we explore combining two pre-trained models from different domains into a sequence-to-sequence framework with the help of adapters.

**Parallel Decoding**    Parallel sequence decoding hugely reduces the inference latency by neglecting the conditional dependency between output tokens, based on novel decoding algorithms including non-autoregressive decoding [9, 11, 32, 12], insertion-based decoding [31, 10] and Mask-Predict [8, 13]. In Mask-Predict, the framework is trained as a conditional masked language model as:

$$L_{\text{CMLM}}(y^m|y^r, x; \theta_{\text{enc}}, \theta_{\text{dec}}) = -\sum_{t=1}^{|y^m|} \log P(y_t^m|y^r, x; \theta_{\text{enc}}, \theta_{\text{dec}}), \tag{2}$$

where $(x, y)$ is a sample of parallel training pairs from the dataset, $y^m$ and $y^r$ are the masked/residual target tokens, $\theta_{\text{enc}}$ and $\theta_{\text{dec}}$ are the parameters of the encoder and decoder respectively. During inference, the model iteratively generates the target sequence in a mask-and-predict manner, which fits well with the bi-directional and conditional independent nature of BERT. Inspired by that, we conduct training and inference of our model in a similar way, which is introduced in Section 3.3.

## 3    Framework

In this section we introduce the proposed framework of fine-tuning BERT with adapters, termed as Adapter-Bert Networks (AB-Net) and illustrated in Figure 1. We start with the problem definition.

**Problem Definition**    Given two pre-trained BERT models XBERT and YBERT on the source side and the target side respectively, we aim at fine-tuning them in a sequence-to-sequence framework by introducing adapter modules, on a parallel training dataset $(\mathcal{X}, \mathcal{Y})$ which consists of pairs of source and target sequences $(x, y) \in (\mathcal{X}, \mathcal{Y})$. The loss function of our framework is defined in a similar way as the conditional MLM loss introduced in Equation (2):

$$L(y^m|y^r, x; \theta_{\text{AENC}}, \theta_{\text{ADEC}}) = -\sum_{t=1}^{|y^m|} \log P(y_t^m|y^r, x; \theta_{\text{AENC}}, \theta_{\text{ADEC}}), \tag{3}$$

where $\theta_{\text{AENC}}$ and $\theta_{\text{ADEC}}$ indicate the parameters of encoder adapters and decoder adapters respectively.

### 3.1    Adapter-Bert Networks

The architecture of BERT [5] is akin to a Transformer encoder [34], which is constructed by self-attention, feed-forward layers, layer normalization [1] and residual connections [14]. We denote a

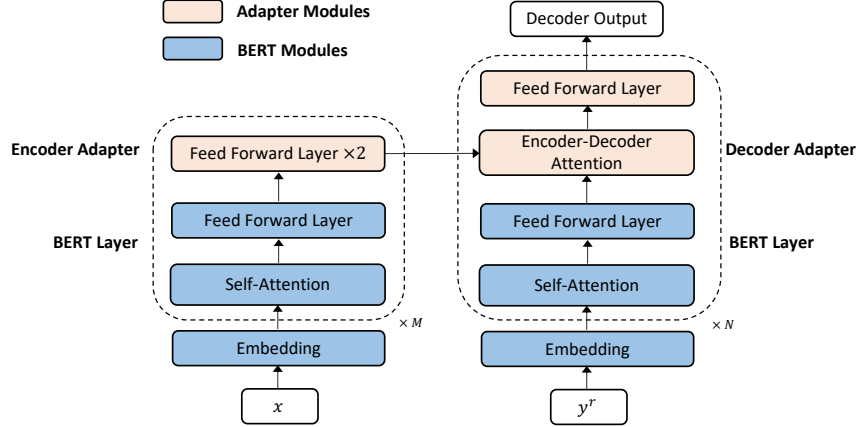

Figure 1: An illustration of the proposed framework. Blue blocks constitute the pre-trained BERT models which are frozen during fine-tuning, and orange blocks represent the adapter components which are inserted into each BERT layer and trained during fine-tuning. $x$ and $y^r$ represent the source sequence and the residual target sequence in Equation (3) respectively. $M$ and $N$ indicate the number of layers of the encoder and decoder. For simplicity, we omit some architecture details such as layer normalization and residual connections.

BERT layer block as $\text{XBERT}(\cdot)$ or $\text{YBERT}(\cdot)$. Please refer to Appendix A for more details about the model architectures.

To fine-tune BERT for natural language generation, we introduce adapter layers and insert them into each BERT layer. On the encoder side, we follow [3] and simply construct the adapter layer with layer normalization as well as two feed-forward networks with non-linearity between them:

$$Z = W_1 \cdot (\text{LN}(H)), \quad H_{\text{AENC}} = H + W_2 \cdot (\text{ReLU}(Z)), \tag{4}$$

where $H$ and $H_{\text{AENC}}$ are the input and output hidden states of the adapter respectively, LN indicates layer normalization, $W_1$ and $W_2$ are the parameters of the feed-forward networks. The only hyper-parameter brought by this module is the dimension $d_{\text{Aenc}}$ of the internal hidden state $Z$, through which we can flexibly control the capacity and efficiency of adapter layers. Denoting the encoder adapter layer as $\text{AENC}(\cdot)$, for each encoder layer in the framework, the hidden state is computed as:

$$H_{l+1}^E = \text{AENC}(\text{XBERT}(H_l^E)), \tag{5}$$

where $H_l^E$ is the output hidden state of the $l$-th encoder layer. And we take the hidden state $H^E$ of the last encoder layer as the representation of the source sequence.

As for the decoder, the introduced adapter modules should be able to model the conditional information from the source side. Therefore, we adopt the multi-head cross-attention computed over the encoder output and decoder input as the adapter. Denote the attention based adapter as $\text{ADEC}(Q, K, V)$, then given the hidden output $H_l^D$ of the $l$-th decoder layer, the hidden state of the $(l+1)$-th layer is calculated as:

$$H_{l+1}^D = \text{ADEC}(\text{YBERT}(H_l^D), H^E, H^E). \tag{6}$$

By introducing and carefully designing the adapter modules on the encoder and decoder, our framework is able to utilize the pre-trained information from both sides as well as build the conditional dependency, making it possible to apply the model on conditional text generation tasks.

## 3.2 Discussion

To the best of our knowledge, our framework is the first work that is able to jointly integrate pre-trained models from both the encoder and decoder sides. Different from most previous works that plainly utilize BERT as a feature extractor [41, 38, 37], we directly exploit BERT as the encoder and decoder to make the most of pre-trained models. Comparing with the related works that also utilize adapter modules while fine-tuning [16, 36, 3], we do not constrain the architectures of adapters to be fixed, but adopt different architectures on the encoder and decoder sides. In addition, we can easily

extend the architectures of adapters to adjust to different downstream tasks. For example, while our framework is designed for parallel decoding, it is straightforward to transform it to traditional autoregressive decoding by extending the cross-attention based adapter to a traditional Transformer decoder. We show in Table 3a that our autoregressive variant is able to achieve strong performance.

Meanwhile, by integrating two large scale pre-trained models into a sequence-to-sequence framework, we have illustrated their benefits as well as drawbacks. The main limitation is the extra computation cost brought by the enormous pre-trained parameters. Fortunately, thanks to the lightweight and flexible adapter modules, the scale of parameters that require training in our framework is less than that of an autoregressive Transformer model. Besides, we have multiple ways to adjust the scale of adapters. For example, while training, instead of inserting adapter layers to all BERT layers, we can only insert them into the top layers to speed up training. We can also reduce the hidden dimensions of adapters to control the parameter scale with negligible degradation of performance. A thorough study is conducted regarding the flexibility of adapters in Section 4.4. It can be shown that, at the inference stage, even with two large pre-trained models introduced, our framework based on parallel decoding can still achieve faster decoding speed than traditional autoregressive baselines.

### 3.3 Training and Inference

We mainly follow the training and inference paradigm used in [8]. To decode the target sequence in parallel, the model needs to predict the target length conditioned on the source sequence, i.e., modeling $P(|y| \, | \, x)$. We add a special [LENGTH] token to the encoder input, and take its encoder output as the representation, based on which the target length is predicted. The length prediction loss is added to the word prediction loss in Equation (3) as the final loss of our framework. In Equation (3), given a training pair $(x, y)$, we randomly mask a set of tokens in $y$ with [MASK], and the number of the masked tokens $|y^m|$ is uniformly sampled from 1 to $|y|$ instead of being computed by a fixed ratio as BERT [5]. The masked tokens are denoted as $y^m$ while the residual tokens are denoted as $y^r$,

While inference, the target sequence is generated iteratively in a mask-and-predict manner. Specifically, after the length of the target sequence is predicted by the encoder, the decoder input is initialized with the [MASK] symbol at all positions. After the prediction process of the decoder, a number of tokens with the lowest probabilities in the decoder output are replaced by [MASK]. The obtained sequence is taken as the decoder input of the next iteration until the stop condition is hit. The number of masked tokens at each iteration follows a linear decay function utilized in [8]. As for the stop condition, the final result is obtained either when the upper bound of iterations is reached, or the obtained target sequence do not change between two consecutive iterations. Details of the decoding algorithm are provided in Appendix B.

## 4 Experiments

We mainly conduct experiments on neural machine translation to evaluate our framework. We also explore its autoregressive variant in Section 4.3, followed with ablation studies in Section 4.4.

### 4.1 Experimental Setup

**Datasets** We evaluate our framework on benchmark datasets including IWSLT14 German→English (IWSLT14 De-En)[2], WMT14 English↔German translation (WMT14 En-De/De-En)[3], and WMT16 Romanian→English (WMT16 Ro-En)[4]. We show the generality of our method on several low-resource datasets including IWSLT14 English↔Italian/Spanish/Dutch (IWSLT14 En↔It/Es/Nl). We additionally consider WMT14 English→French translation (WMT14 En-Fr) for autoregressive decoding. We follow the dataset configurations of previous works strictly. For IWSLT14 tasks, we adopt the official split of train/valid/test sets. For WMT14 tasks, we utilize `newstest2013` and `newstest2014` as the validation and test set respectively. For WMT16 tasks, we use `newsdev2016` and `newstest2016` as the validation and test set. For autoregressive decoding, we consider WMT16 Ro-En augmented with back translation data[5] to keep consistency with base-

Table 1: The BLEU scores of the proposed AB-Net and the baseline methods on the IWSLT14 De-En, WMT16 Ro-En and WMT14 En-De/De-En tasks. The per-sentence decoding latency and the number of trained parameters on the WMT14 En-De task are also reported. "∗" indicates the results obtained by our implementation, "/" indicates the corresponding result is not provided.

| | IWSLT14 | WMT16 | WMT14 | | | #Trained |
| Models | De−En | Ro−En | En−De | De−En | Latency | Parameters |
|---|---|---|---|---|---|---|
| Transformer-Base [34] | 33.59* | 34.46* | 28.04* | 32.69* | 778* ms | 74M |
| Mask-Predict [8] | 31.71* | 33.31 | 27.03 | 30.53 | 161* ms | 75M |
| BERT-Fused NAT [41] | 33.14* | 34.12* | 27.73* | 32.10* | 260* ms | 90M |
| **AB-Net** | **36.49** | **35.63** | **28.69** | **33.57** | 327 ms | 67M |
| **AB-Net-Enc** | 34.45 | / | 28.08 | / | 165 ms | 78M |

lines [41]. We tokenize and segment each word into wordpiece tokens with the internal preprocessing code in BERT[6] using the same vocabulary as pre-trained BERT models, resulting in vocabularies with 30k tokens for each language. More details of the datasets are described in Appendix C.

**Model Configurations** We mainly build our framework on `bert-base` models ($n_{layers} = 12$, $n_{heads} = 12$, $d_{hidden} = 768$, $d_{FFN} = 3072$). Specifically, for English we use `bert-base-uncased` on IWSLT14 and `bert-base-cased` on WMT tasks. We use `bert-base-german-cased` for German and `bert-base-multilingual-cased` for all other languages. When extending to autoregressive decoding, we utilize `bert-large-cased` ($n_{layers} = 24$, $n_{heads} = 16$, $d_{hidden} = 1024$, $d_{FFN} = 4096$) for English to keep consistency with [41]. For adapters, on the encoder side, we set the hidden dimension between two FFN layers as $d_{Aenc} = 2048$ for WMT tasks and 512 for IWSLT14 tasks. On the decoder side, the hidden dimension of the cross-attention module is set equal to the hidden dimension of BERT models, i.e., $d_{Adec} = 768$ for `bert-base` models and $d_{Adec} = 1024$ for `bert-large` models. We train our framework on 1/8 Nvidia 1080Ti GPUs for IWSLT14/WMT tasks, and it takes 1/7 days to finish training. Our implementation is based on `fairseq` and is available at `https://github.com/lemmonation/abnet`.

**Baselines** We denote the proposed framework as AB-Net, and to make a fair comparison with baseline models, we also consider a variant of our model that only incorporates BERT on the source-side with encoder adapter layers and denote it as AB-Net-Enc. With parallel decoding, we consider Mask-Predict [8] as the backbone training and inference algorithm, based on which we re-implement BERT-Fused [41] and take it as the main baseline, denoted as BERT-Fused NAT. With autoregressive decoding where BERT is utilized only on the source-side, we compare our framework with BERT-Fused [41], BERT-Distilled [4] and CT-NMT [38] with their reported scores.

**Inference and Evaluation** For parallel decoding, we utilize sequence-level knowledge distillation [18] on the training set of WMT14 En-De/De-En tasks, to keep consistency with [8]. This technique has been proved by previous non-autoregressive models that it can produce less noisy and more deterministic training data [9]. We use the raw training data for all other tasks. While inference, we generate multiple translation candidates by taking the top $B$ length predictions into consideration, and select the translation with the highest probability as the final result. We set $B = 4$ for all tasks. And the upper bound of iterative decoding is set to 10. For autoregressive decoding, we use beam search with width 5 for all tasks. We utilize BLEU scores [25] as the evaluation metric. Specifically, we use `multi-bleu.perl` and report the tokenized case-insensitive scores for IWSLT14 tasks and tokenized case-sensitive scores for WMT tasks.

## 4.2 Results

The results of our framework with parallel decoding are listed in Table 1. The autoregressive Transformer model with `base` configuration [34] is also compared as a baseline. In addition to BLEU scores, we also report the per-sentence decoding latency on the `newstest2014` test set as well as the number of trained parameters on the WMT14 En-De task. As can be observed from Table 1,

Table 2: The performance of the proposed AB-Net on IWSLT14 low-resource language pairs. Mask-Predict as well as the autoregressive Transformer-Base model are considered as baselines.

| Models | En-It | It-En | En-Es | Es-En | En-Nl | Nl-En |
|---|---|---|---|---|---|---|
| Transformer-Base [34] | 29.26 | 33.57 | 36.04 | 39.31 | 31.30 | 36.19 |
| Mask-Predict [8] | 26.05 | 29.50 | 32.15 | 35.37 | 25.78 | 32.91 |
| **AB-Net** | **31.81** | **34.20** | **37.45** | **42.66** | **32.52** | **38.94** |

Table 3: (a) The results of machine translation with autoregressive decoding of our framework and baseline methods. We directly copy the best results reported in their papers. Transformer-Big indicates Transformer with `transformer-big` configuration. (b) The ablation study on different components of the proposed model conducted on the test set of IWSLT14 De-En. "Decoder" indicates a traditional Transformer decoder. $\times$ indicates the setting with no convergence reached during training.

(a) Results on autoregressive decoding.

| Models | WMT14 En$-$De | WMT14 En$-$Fr | WMT16 Ro$-$En |
|---|---|---|---|
| Transformer-Big [34] | 28.91* | 42.23* | 36.46* |
| BERT-Distilled [4] | 27.53 | / | / |
| CT-NMT [38] | 30.10 | 42.30 | / |
| BERT-Fused [41] | 30.75 | 43.78 | 39.10 |
| AB-Net-Enc | 30.60 | 43.56 | 39.21 |

(b) Ablation study.

| Model Variants | BLEU |
|---|---|
| Transformer-Base | 33.59 |
| (1): X$_{\text{BERT}}$ + Decoder | $\times$ |
| (2): (1) + A$_{\text{ENC}}$ | 34.45 |
| (3): (2) + Y$_{\text{BERT}}$ | $\times$ |
| (4): (3) + A$_{\text{DEC}}$ | 36.49 |
| (5): (4) + A$_{\text{DEC}}$ on top 6 layers | 34.60 |
| (6): (5) + A$_{\text{ENC}}$ on top 6 layers | 33.78 |

with parallel decoding, Mask-Predict achieves considerable inference speedup but also suffers from performance degradation at the same time. Equipped with pre-trained BERT models from both sides, our framework obtains a huge performance promotion compared with Mask-Predict. In addition, we also outperforms the autoregressive baseline Transformer-Base by a firm margin with similar scales of trainable parameters, while achieving 2.38 times speedup regarding the inference speed. Compared with BERT-Fused NAT [41] which utilizes BERT only on the encoder side, our framework as well as the BERT-encoder-only variant AB-Net-Enc both achieve better performance with less parameters to train, illustrating that the introduced adapter modules are able to leverage more information in a more efficient way.

Regarding the scale of trained parameters, we can notice that AB-Net-Enc actually introduces more parameters than AB-Net which utilizes BERT from both sides. The reason lies in the embedding layer. By incorporating BERT through adapters, the proposed framework gets rid of training the giant embedding layer which usually introduces ~15M parameters to train on each side if embeddings are not shared. Comparing with BERT-Fused NAT [41], our framework is able to save ~26% parameters while incorporating information from both sides, providing a more cost-effective solution for leveraging pre-trained models based on adapters.

**Results on Low-Resource Language Pairs** We also study the performance of our framework on three low-resource language pairs in the IWSLT14 dataset. Results on both directions are shown in Table 2. The proposed AB-Net consistently outperforms the compared baselines among various language pairs, demonstrating the generality of our method.

## 4.3 Exploration on Autoregressive Decoding

Here we explore the application of our framework on autoregressive decoding. As the bidirectional and conditional independent nature of BERT prevents it from being applied to autoregressive decoding, to show the flexibility of the proposed framework, we directly use AB-Net-Enc as the autoregressive variant, whose encoder is initialized with the source-side BERT model and equipped with encoder adapter layers, while the decoder is an autoregressive Transformer Decoder. We compare our model with three fine-tuning baselines including BERT-Fused [41], BERT-Distilled [4] and CT-NMT [38]. Results are shown in Table 3a. Our framework outperforms the Transformer-Big baseline over all three translation tasks, with improvements from 1.33 to 2.75 BLEU scores. BERT-Fused [41]

also achieves considerable performance. Nevertheless it is worth noting that BERT-Fused requires pre-training a standard Transformer model at first (without which there will be a drop of $2.5$ BLEU score as reported), which is time consuming. While we simply train our Transformer decoder from scratch, we expect additional performance gains if similar tricks are applied.

We have also explored other alternatives. In practice, the bi-directional property is achieved by setting the attention mask as a matrix with all 1s to enable each token to see all other tokens. Therefore, in our framework, we try to fine-tune BERT on autoregressive decoding by setting the attention matrix of the decoder as an upper triangle matrix to prevent the model from attending to the future words. However, in this way, we can only achieve sub-optimal results compared with the BERT encoder variant AB-Net-Enc ($26.40$ vs $29.96$ on the test set of IWSLT14 En-De). One possible reason is that the introduced adapter parameters are not powerful enough to change the bidirectional nature of BERT without tuning it. Another solution is to mingle autoregressive pre-trained models with BERT to construct a hybrid framework, e.g., a BERT encoder and a GPT decoder, which fits well in nature with an autoregressive sequence-to-sequence model. We leave that for future work.

## 4.4 Ablation Study

In this subsection, we further conduct ablation studies regarding the scale of adapters, the proposed different components, different fine-tuning strategies and baselines with back-translation. Experiments are conducted on the IWSLT14 De-En dataset with parallel decoding.

**Ablations on the Scale of Adapters**  We investigate the influence of the scale of adapters in Figure 2. Specifically, we fix the scale of the decoder adapter and tune the hidden dimension of the encoder adapter $d_{\text{Aenc}}$ in a wide range ($2^6$ to $2^{10}$). We also plot the number of trained parameters on the encoder side in our framework and in the Transformer-Base model to make a comparison. From Figure 2, we find that our framework is robust to the scale of adapters, e.g., halving the dimension from $2^9$ to $2^8$ only results in a $0.4$ drop of the BLEU score. Compared with the autoregressive Transformer baseline, our framework is able to achieve better performance with only $5\%$ parameters to train (getting $34.81$ score when $d_{\text{Aenc}} = 64$), illustrating the efficiency of adapters.

**Ablations on the Different Proposed Components**  In Table 3b, we study the influence of different components in our framework including the pre-trained BERT models (XBERT and YBERT) and adapter layers (AENC and ADEC).

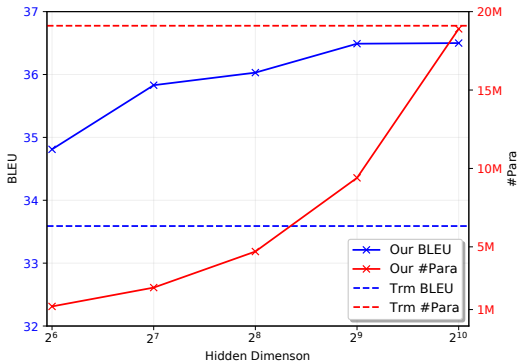

Figure 2: The study on the scale of encoder adapters. Blue lines with the left y-axis indicate BLEU scores while red lines with the right y-axis indicate the number of parameters to train on the encoder side. Trm indicates the Transformer-Base model. Best view in color.

We can find that without utilizing adapters on either side, the model cannot converge during training, indicating the necessity of the adapter modules. While BERT models are usually very deep (12 layers), we also explore to insert adapters into top layers only (i.e., 7~12-th layers) to reduce the scale of the introduced parameters in settings (5) and (6). As shown in Table 3b, when only introducing adapters to the top layers of both XBERT and YBERT following setting (6), our framework still outperforms the autoregressive Transformer baseline, showing that it is flexible to balance the performance and scale of our framework. In addition, we can find that the decoder adapter contributes more than the encoder adapter, illustrating the importance of modeling the conditional dependency over all scales of hidden representations.

**Comparison with Different Fine-Tuning Strategies**  In addition to the proposed model which freeze the BERT components and only tune the adapters while training, we also consider the variant that fine-tunes the full model in AB-Net (AB-Net FB), or trains AB-Net from scratch (AB-Net SC). We train all variants for 50 epochs and evaluate on the validation set of IWSLT14 De-En. Results are shown in Figure 3a, where AB-Net converges significantly faster than AB-Net FB, and AB-Net SC does not converge. When fine-tuning the full model, more GPU memory is required because

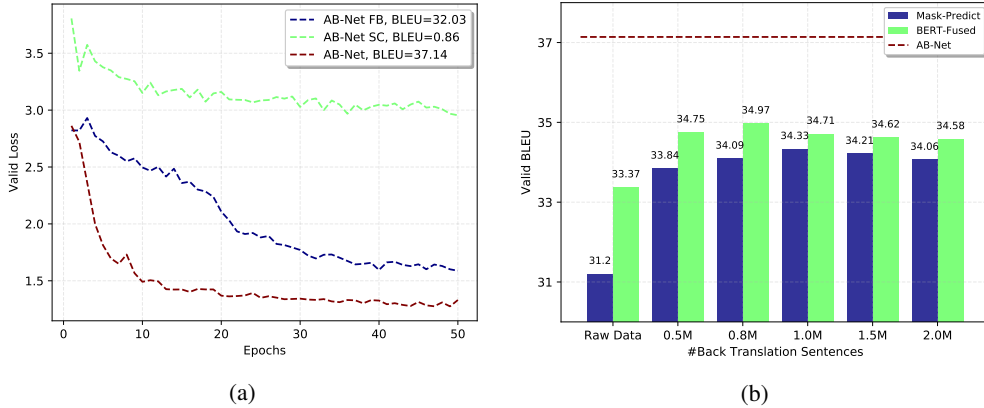

|       |       |
|-------|-------|
| (a)   | (b)   |

Figure 3: (a): Results of different fine-tuning strategies. (b): Results of baselines trained with extra monolingual data via back-translation. All settings are evaluated on the validation set of the IWSLT14 De-En task.

more gradient states need to be stored, therefore we have to halve the batchsize to fit the model into GPUs, which slows down the training process. With the same batchsize, AB-Net saves 29% GPU memory and 26% wall-clock training time compared with AB-Net FB. Moreover, we find that directly fine-tuning BERT is very unstable and sensitive to the learning rate, while only tuning the adapters alleviates this problem and is relatively more robust.

**Comparison with Back-Translation** Back-translation is a simple yet effective data augmentation method in NMT [7, 40]. While we leverage BERT models which are pre-trained with extra monolingual data, we also consider baselines trained with extra monolingual data via back-translation to construct fair comparisons. Specifically, we first train an AT model on the IWSLT14 En-De task (with a 28.96 BLEU score), and then use it to generate additional training pairs on the English Wikipedia data, which is a subset of the training corpus of BERT. Results are shown in Figure 3b. We can find that the gains brought by back-translation are limited, and adding over 1M monolingual data actually brings a performance drop. In addition, comparing with our method, back-translation requires to train another model and decode a large amount of monolingual data, which is time consuming.

## 5 Conclusion

In this paper, we propose a new paradigm of incorporating BERT into text generation tasks with a sequence-to-sequence framework. We initialize the encoder/decoder with a pre-trained BERT model on the source/target side, and insert adapter layers into each BERT layer. While fine-tuning on downstream tasks, we freeze the BERT models and only train the adapters, achieving a flexible and efficient framework. We build our framework on a parallel decoding method named Mask-Predict to match the bidirectional and conditional independent nature of BERT, and extend it to traditional autoregressive decoding in a straightforward way. Our framework avoids the catastrophic forgetting problem and is robust when fine-tuning pre-trained language models. The framework achieves strong performance on neural machine translation while doubling the decoding speed of the Transformer baseline. In the future, we will try to combine two different pre-trained models together in our framework, such as a BERT encoder and a GPT/XLNet decoder, to explore more possibilities on autoregressive decoding.

## Acknowledgements

This research was supported by the National Natural Science Foundation of China (61673364, U1605251), Anhui Provincial Natural Science Foundation (2008085J31) and the National Key R&D Program of China (2018YFB1403202). We would like to thank the Information Science Laboratory Center of USTC for the hardware and software services. We thank the anonymous reviewers for helpful feedback on early versions of this work. The work was done when the first author was an intern at Alibaba.

## Broader Impact

The proposed framework can be seen as a new and general paradigm of designing sequence-to-sequence models by leveraging pre-trained models, which has a wide range of applications not limited to the text generation tasks discussed in the paper, e.g., end-to-end speech/image translation. If proper pre-trained models for the specific task are provided (such as BERT for text or ResNet for images), our framework can then provide a cost-effective solution to leverage them without tuning their massive parameters or re-training them from scratch, which will save lots of resources for researchers who are individual or affiliated with academic institutions. In addition, different from most pre-training approaches which particularly focus on English, our framework works well when dealing with various low-resource languages as shown in the paper, therefore we may help improve the performance of existing low-resource machine translation systems.

On the other hand, although we only need to tune the adapters while training, we have to load the whole framework into GPUs, which limits the choices of hyper-parameters because large-scale BERT models will occupy more memory than traditional Transformer models. As a consequence, our framework may not perform as its best on GPUs with limited memory. From a broader perspective, the proposed framework is not free from the risks of automation methods. For example, the model may inherit the biases contained in data. And in our framework, both pre-training and fine-tuning datasets may have biases. Therefore we encourage future works to study how to detect and mitigate similar risks that may arise in our framework.

## Footnotes

[2] `https://wit3.fbk.eu/`

[3] `https://www.statmt.org/wmt14/translation-task`

[4] `https://www.statmt.org/wmt16/translation-task`

[5] `http://data.statmt.org/rsennrich/wmt16_backtranslations/ro-en`

[6]`https://github.com/huggingface/transformers/blob/master/src/transformers/tokenization_bert.py`

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
