[Supplementary Material]

# A  The Architecture of Decoder Adapters

We mainly follow [34] to design our attention based adapter on the decoder side. Specifically, the decoder adapter $\text{ADEC}(Q, K, V)$ consists of the attention module, feed-forward layers, layer normalization and residual connections, where $Q, K, V$ indicate the query, key and value matrices respectively. The attention module is computed as follows,

$$\text{ATTN}(Q, K, V) = \text{softmax}(\frac{QK^T}{\sqrt{d_k}})V,$$

where $d_k$ is the hidden dimension of the key matrix $K$. We also follow [34] and implement the multi-head version of the attention module, and please refer to [34] for the details. In our framework, the query vector is from the decoder side (denoted as $H_l^D$) while the key and value vectors are both from the encoder side (denoted as $H^E$). In our experiments, the hidden dimension of encoder and decoder representations are the same, therefore we have $d_q = d_k = d_v = d_{\text{Adec}}$.

Following the attention layer are the feed-forward layers:

$$\text{FFN}(H) = \text{ReLU}(HW_1 + b_1) \cdot W_2 + b_2,$$

where $W_1 \in \mathbb{R}^{d_{\text{Adec}} \times d_{\text{FFN}}}, W_2 \in \mathbb{R}^{d_{\text{FFN}} \times d_{\text{Adec}}}, b_1 \in \mathbb{R}^{d_{\text{FFN}}}, b_2 \in \mathbb{R}^{d_{\text{Adec}}}$ are the parameters to learn in the FFN layers, and the internal dimension $d_{\text{FFN}}$ is set to be consistent with the Transformer baseline. Specifically, when considering Transformer-Base or Transformer-Big as baselines, we set $d_{\text{FFN}} = 2048$ or $d_{\text{FFN}} = 4096$ to be consistent with the `transformer-base` or `transformer-big` condigurations.

Along with layer normalization (LN) and residual connections, the computation flow of the proposed decoder adapter can be written as:

$$Z = \text{LN}\left(\text{ATTN}(\text{YBERT}(H_l^D), H^E, H^E) + \text{YBERT}(H_l^D)\right),$$
$$H_{l+1}^D = \text{LN}\left(\text{FFN}(Z) + Z\right),$$

which is denoted as $\text{ADEC}(Q, K, V)$ in the main content.

# B  Decoding Algorithm

While decoding, we follow [8] and utilize a linear decay function to decide the number of masked tokens in each iteration:

$$|y^m| = \lfloor |y| \cdot \frac{T - t}{T} \rfloor,$$

where $\lfloor \cdot \rfloor$ indicates the floor function, and $T$ is the upper bound of the iteration times while $t$ indicates the number of the current iteration. We set $T = 10$ over all tasks, therefore after the initial iteration when all positions are predicted, we will then mask $90\%, 80\%, ..., 10\%$ tokens in following iterations. And for each iteration, we only update the probabilities of masked tokens while keeping the probabilities of unmasked tokens unchanged. In Table 4 we provide an illustration of the decoding process of our model.

In the main content, we also report the inference latency of different models in Table 1. Specifically, we set the batch size to 1 while inference and calculate the average per sentence translation time on `newstest2014` for the WMT14 En-De task. We run all models on a single Nvidia 1080Ti GPU for a fair comparison.

# C  Detailed Experimental Settings

We list the statistics of datasets utilized in the neural machine translation tasks in Table 5. For IWSLT14 tasks, we follow the preprocessing script provided in `fairseq`[7]. Specifically, we concat `dev2010`, `dev2012`, `tst2010`, `tst2011` and `tst2012` as the text set for each task, and the validation set is split from the training set. For WMT tasks, we follow the dataset settings described in the main content.

Table 4: An illustration of our model with different number of decoding iterations $k$ on the test set of the IWSLT14 De-En task. The underlined words indicate the masked words in the next iteration. "##" is the segment symbol of wordpiece tokens.

| Source: | oder das , was ich mir heute vors ##tell , weil was sie sich gedacht haben könnten . |
|---|---|
| Target: | or anything that i imagine because they might have thought . |

*AB-Net with different iteration $k$*

| $k = 1$: | or what i i imagine imagine today because what you might have thought . |
|---|---|
| $k = 2$: | or maybe what i i imagine today because what you might have thought . |
| $k = 3$: | or maybe the what i imagine today because what you might have thought . |
| $k = 4$: | or it is what i imagine today because what you might have thought . |

Table 5: Dataset Statistics

|  | IWSLT14 | | | | WMT14 | | WMT16 | |
|---|---|---|---|---|---|---|---|---|
|  | De-En | En↔It | En↔Es | En↔Nl | En↔De | En→Fr | Ro→En | Ro→En + BP |
| #Train | 157k | 167k | 169k | 154k | 4.5M | 36M | 610k | 2.6M |
| #Valid | 7k | 8k | 8k | 7k | 3k | 3k | 2k | 2k |
| #Test | 7k | 6k | 6k | 5k | 3k | 3k | 2k | 2k |

While preprocessing, we use the same vocabulary of BERT models to decode the dataset. As introduced in the main content, with parallel decoding, we use `bert-base-uncased`/`bert-base-cased` (on IWSLT14/WMT tasks) for English, `bert-base-german-cased` for German and `bert-base-multilingual-cased` for other languages. With autoregressive decoding, we use `bert-large-cased` for English. Specifically, `bert-base-uncased`/`bert-base-cased`/`bert-base-german-cased` are equipped with vocabularies containing 30k/29k/30k tokens, while the dictionary of `bert-base-multilingual-cased` contains 119k tokens, which is much larger because it consists of the common tokens among 104 languages. For each low-resource language considered in our experiments, directly loading the whole embedding matrix of the multilingual BERT model will waste a lot of GPU memory. Therefore we only consider tokens that appear in the training and validation set, and manually modify the checkpoint of the multilingual BERT to omit the embeddings of unused tokens. In this way, we obtain dictionaries that contain 24k/16k/17k/16k tokens for Ro/It/Es/Nl respectively, which ultimately save around 77M parameters in average.

The links to download the BERT models utilized in our paper are listed below:

- `bert-base-uncased`: https://s3.amazonaws.com/models.huggingface.co/bert/bert-base-uncased.tar.gz

- `bert-base-cased`: https://s3.amazonaws.com/models.huggingface.co/bert/bert-base-cased.tar.gz

- `bert-large-cased`: https://s3.amazonaws.com/models.huggingface.co/bert/bert-large-cased.tar.gz

- `bert-base-german-cased`: https://int-deepset-models-bert.s3.eu-central-1.amazonaws.com/pytorch/bert-base-german-cased.tar.gz

- `bert-base-multilingual-cased`: https://s3.amazonaws.com/models.huggingface.co/bert/bert-base-multilingual-cased.tar.gz

## Footnotes

[7] https://github.com/pytorch/fairseq/blob/master/examples/translation/ prepare-iwslt14.sh