[Reviews · NeurIPS 2020]

Review 1

Summary and Contributions: This work uses pretrained source and target LMs to improve machine translation. In the non-autoregressive setting, they use BERTs with mask-predict (to match the pretraining condition). To speed up training they use adapters, in particular a cross-attention adapter for the decoder. This yields consistent improvements in BLEU over NAR baselines without pretraining, and over other MLM+MT integrations. Ablations for adapters and encoder vs. decoder included. ---- AFTER AUTHOR FEEDBACK: Thanks for showing improvements in initial convergence and quantifying savings in memory and time. Still, consider explaining why # of trained parameters remains so high in the text. While I'd like to see control on extra pretraining data, the backtranslation experiments ablate the approach's contribution in a similar way. My concern re IWSLT'14 BLEU was a misunderstanding (Zhu used german-cased despite the lowercased setting). I've increased my score. HOWEVER, >>the Mask-Predict numbers for WMT'16 Ro-En seem to be lowercase BLEU<<. This ACL'20 paper (Footnote 3: https://www.aclweb.org/anthology/2020.acl-main.171.pdf) suggests case-insensitive is standard for NAT, at least for approaches that follow Lee et al. (iterative refinement) and Mask-Predict (see data in https://github.com/facebookresearch/Mask-Predict/blob/master/get_data.sh). Zhu was AT and used cased, and so do other NAT works (https://arxiv.org/pdf/1711.02281.pdf, https://arxiv.org/abs/1812.09664). Hence, either use uncased to compare w/ Mask-Predict, or re-evaluate Mask-Predict using cased. Sorry for this annoying situation.

Strengths: - Principled integration of known methods (mask-predict + MLMs, adapters + cross-attention) to an important problem (using pre-trained models for MT). - Interesting ablations (adapter scale, decoder vs. encoder adapters vs. which layers)

Weaknesses: - Methods are known individually; only synthesis is new - Needs more analysis of key selling points: (i) gains from data versus from MLM/TLM pretraining + examples of translation using BERT's knowledge; (ii) training speedup from adapters.

Correctness: Methodology, results, and comparisons are proper. - L230: Expand this (here or appendix). Can you give SacreBLEU tag or script names? Tokenized and case-insensitive BLEU can appear non-standard. Explain that it's because of prior work (Zhu [40] seems to use IWSLT'14 cased though). Is this because you used BERT uncased? Why (does BERT cased not work well)?

Clarity: Paper is well-written. - A few more sentences on how catastrophic forgetting manifests (either L37, L59, or in Sec 2) when you swap encoders directly will help readers. - Sec 3.3, etc.: Reference your Appendix, which has great details! - Table 3b and 303-314: I might not understand, but the description for (5) and (6) makes it sound like you should write "(2) + ..." instead of "(4/5) + ...".

Relation to Prior Work: The novelties over previous MLM + MT and adapter uses are clear. Further references below but I leave at author's discretion: L95-101: Other MLM + MT integrations: * MLM encoders directly (not KD): https://www.aclweb.org/anthology/D19-5611.pdf, https://www.aclweb.org/anthology/D19-5603. Acknowledge that XLM and Zhu did encoder and decoder MLM pretraining also (L87: not necessarily parallel either, only TLM did that) * MLMs via score interpolation: https://arxiv.org/abs/1910.14659 (and recently https://arxiv.org/abs/2004.08097). - L105-107: Description should be fixed. The insertion models aren't (all) parallel; they change generation order. Mask-Predict being exclusive from "NAR decoding" is debatable. Perhaps write "non-autoregressive decoding [8, 31] with refinement [https://www.aclweb.org/anthology/D18-1149/, 7]. Our method of interest is Mask-Predict [7]..." - L242: Take credit! Only BERT-fused is from Zhu [40]; make clear the NAT part is from you(?) - L272-279: https://arxiv.org/abs/1904.09408 also considered finetuning BERT to be unidirectional.

Reproducibility: Yes

Additional Feedback: L1, L24, L74, etc.: "pre-training language models" --> "pre-trained language models" L92: "corpuses" --> "corpora" L108: "between tokens" --> "between output tokens" L112, L178: "While inference" --> "During inference" Figure 1: "freezed" --> "frozen". Clarify what happens to $y_r$ at inference. Table 1: Explain "AB-Net-Enc" in the caption (doesn't get explained until mid next page). Q: Did you ablate more pretraining size (e.g., BERT vs. RoBERTa)? Conversely, less data: do MLM on the WMT corpus only, then use adapters. "How much" of the gain is from added data? E.g., qualitative examples (translations showing BERT's knowledge). Q: How much faster was training with adapters (encoder-only adapters, last 6 layers only)? This is a key motivation, so we should see results, e.g. its own paragraph / table / plots (otherwise Table 1 makes you parameter savings look small).


Review 2

Summary and Contributions: The paper introduces a framework based on 'adapters' that allows you to combine two pre-trained BERT models and train them on non-autoregressive MT (NAT) with the mask-predict method. The main novelty is the use of encoder-decoder attention in adapters and the task (NAT) that they are applied to.

Strengths: It's a nice, simple method, that produces strong results. It could be useful to the community to see pre-trained models used in a novel way, and increases NAT performance 'for free', as long as you have the correct pre-trained model.

Weaknesses: There is a lack of novelty (the components of the method are well known), but I don't consider this a big weakness given the novel combination and extension of previous ideas. There is additionally a lack of a compelling story as to why this method is better than alternatives (see my suggestions of missing baselines further down in this review). For example, if we think the method improves low-resource translation, there is a lack of compelling evidence on distant language pairs (like the FLORES benchmark). If we think the method improves high-resource translation, we should compare to larger models trained from scratch. It could be the model is faster/easier to train than alternatives, but I don't see evidence for that in the paper. Update: My concerns re backtranslation were addressed in the author response. I still strongly encourage the authors to try distant language pairs like My-En etc.!!

Correctness: Yes I don't see any obvious mistakes.

Clarity: Yes, outside of a few minor typos, and perhaps some key information put in the supplementary rather than the main paper.

Relation to Prior Work: I don't see any big omissions, I would additionally cite https://arxiv.org/abs/1908.06938 for adapting pre-trained models to sequence generation and https://arxiv.org/abs/1902.02671 for adapter modules.

Reproducibility: Yes

Additional Feedback: From supplementary: 'we only consider tokens that appear in the training and validation set, and manually modify the checkpoint of the multilingual BERT to omit the embeddings of unused tokens' This is an interesting detail, I think it should be included in the main paper. In equation 5, you pass the encoder layer output directly into an adapter, and the adapter immediately applies layernorm - the final stage of a BERT transformer layer is a residual connection followed by layernorm, so why do we need to apply *another* layernorm in the adapter? The method of Houslby et al. 2019 (and others), applying adapters *before* layernorm is much more intuitive to me. I think you are missing some key baselines. Firstly, large-scale pretraining could be viewed as a form of data augmentation, and so you should compare to normal MT data augmentation, most notably back-translation, which is key to many strong results in MT (e.g. https://arxiv.org/abs/1808.09381). Of course the 'plug and play' nature of using pre-trained models has advantages, since you don't need to generate backtranslation data. However if you are using your method on truly low resource languages (e.g. those in the FLORES benchmark; https://arxiv.org/abs/1902.01382), you may not have a pre-trained model in that language, meaning we should compare the effort needed to do backtranslation vs. pre-training. Additionally applying the method to e.g. Nepali-English or Myanmar-English, or other distantly related languages would be interesting. Another baseline I think is missing is simply training large mask-predict models, from scratch. The NAT models you compare to are only 75m parameters, what happens if you train larger models (with the same number of parameters as AB-net) from scratch? These models may take longer to train due to the larger number of trainable parameters, but then you can compare the AB-net to 'from scratch' at different amounts of wallclock time to show the difference. An optional experiment to try would be to unfreeze certain BERT parameters, say one layer, or only feed-forward layers etc. An empirical study in a different setting is: https://arxiv.org/abs/2004.14911 Typos: l30: 'most successful model' -> 'most successful models' l57 'achieves the following superiority.' -> 'achieves the following benefits.' l105 ' we explore to combine' -> 'we explore combining' l159 'we have obtained their benefits as well as drawbacks' -> 'we have illustrated their benefits as well as drawbacks' l225 'While inference' -> 'During inference' Update: Given most of my concerns were adressed, I've increased my score.


Review 3

Summary and Contributions: Summary This paper proposes a method to integrate pre-trained language models into sequence-to-sequence models. Fine-tuning pre-trained models towards a complicated task such as machine translation results in catastrophic forgetting, thus failing to capitalize on the generalization power of the pre-trained model. The authors froze the original BERT and inserted light weight neural networks in the encoder and decoder layers to decouple task-specific adapters from the pre-trained model. Contributions Novel approach to integrate pre-trained model into sequence-to-sequence by decoupling parameters of the adapters from the pre-trained BERT models The adapters are light-weight and easy to tune The approach is task-agnostic and can be easily integrated flexibly into different problems. For example the authors showed they can make the decoder auto-regressive by just applying their method on the encoder side. The method was evaluated comprehensively across different language pairs and data regimes from low resource to large scale and the efficacy of the method was demonstrated

Strengths: 1) See Contributions in the text-box above 2) The approach is light weight with half of the inference latency.

Weaknesses: Using pre-trained model incur extra computation cost

Correctness: The approach seems sound and the claims made in the paper are backed up with empirical evidence and ablation studies.

Clarity: The paper is clearly written and mostly understandable

Relation to Prior Work: The authors have mentioned previous work on which they build upon.

Reproducibility: No

Additional Feedback:


Review 4

Summary and Contributions: To leverage the source-side and target-side monolingual information, the authors fine-tune two different BERT models as the encoder and decoder by introducing the simple and lightweight adapter modules. Considering the bi-directional and conditional independent nature of BERT, the proposed model is based on the mask-predict and also could be adapted into the autoregressive decoding. The experiments show that the proposed method consistently outperforms autoregressive baselines with twofold decoding speed.

Strengths: 1. The authors design an ingenious adapter to bridge the BERT model on the source-side and the BERT model on the target-side, and can employ the source-side monolingual dataset and the target-side monolingual dataset. The idea is intuitionistic and novel, it can be treated as an integration of previous work related to the work either only using BERT model trained on the source-side data or the target-side data. 2. The model is clear, simple, and the performance on some translation benchmarks are strong comparing to the related works and the strong baselines. 3. The work is very relevant to the NerualPS community, will also be helpful to the community. 4. When applying on the mask-predict decoding algorithm, the decoding speed is almost doubled.

Weaknesses: 1. Although only the adapters need to be fine-tuned during training, the whole framework has to be loaded into GPUs. The selection of hyper-parameters is limited because large-scale BERT models will occupy more memory than traditional Transformer models. 2. For lots of language-pairs, the BERT pretrained models do not exist. I don't know how will the authors deal with this situation. 3. Although the models proposed in the paper greatly exceed the baseline and the related systems on multiple datasets, I think most of the comparisons are unfair. For example, in Table 2, compared with mask-predict and transformer-base, the proposed model leverage the large source-side and target-side monolingual resources, while mask-predict and transformer-base systems do not use these resources. Therefore, the author should give a detailed explanation of the data resources used.

Correctness: The idea is well motivated and reasonable, and the empirical methodology is novel and also correct in the general direction. The experimental design make sense, and the experimental results are convincing to draw conclusions. But there are many experimental comparisons that are not fair.

Clarity: The paper is clearly structured, well and natively written.

Relation to Prior Work: The authors introduce lots of related works detailedly in the paper, and also explain in detail the differences from the relevant methods, such as XLM, MLM, MASS and BART. In addition, the authors also elaborated and compared the advantages and disadvantages of the previous works, such as BERT-fused NMT, BERT-Distilled, Mask-Predict.

Reproducibility: Yes

Additional Feedback: 1. Figure 1 is not mentioned in the paper. 2. For the caption of Figure 1, “the masked target sequence” should be modified into “the residual target tokens” right? 3. Line 173: Could you give the calculation of the length prediction loss? 4. Line 180-181: All positions are initialized with the [MASK] symbol, why do the tokens with the lowest probabilities in the decoder output are still replaced by [MASK] during prediction? 5. As shown in Table 1, comparing to the BERT-Fused NAT, AB-Net-Enc preforms better, which makes sense. Why do you only make comparison on the non-autoregressive decoding? However, I also want to see the comparison between the AB-Net-Enc on autoregressive decoding and the BERT-Fused (which is the 36.11 on the IWSLT14 de-en). Actually, it is a little bit worse than the BERT-Fused model on autoregressive decoding. The authors argue that BERT-Fused requires pre-training a standard Transformer model at first, which is time consuming. Can the similar tricks be applied into the proposed adapters? 6. I hope to see the experimental results when the multilingual BERT pretrained model (such as bert-base-multilingual-cased) is applied. 7. Why are the parameters of BERT layers freezed and only the adapter layers fine-tuned? If the parameters of the BERT model and the parameters of the adapter model are jointly optimized, what will be the result?

[Author Response · NeurIPS 2020]



**Figure 1:** Left: results of different fine-tuning baselines. Right: Results of utilizing back-translation on baselines.

**Common Response**: We would like to thank all reviewers very much for the detailed feedback and valuable suggestions! We will follow the suggestions on writing and related works and revise accordingly. We agree with the reviewers' concerns about the settings of baselines, therefore we provide a common response here, which will be added to the main paper in the revision. We add a new investigation, where we consider the variant of fine-tuning the whole model in AB-Net (AB-Net FB), the variant that trains AB-Net from scratch (AB-Net SC) and baselines trained with back-translation. We train all settings for 50 epochs and test on the validation set of IWSLT14 De-En. For back-translation (BT), we first train an AT model on IWSLT14 En-De (BLEU score 28.96), and then use it to generate additional training pairs on the English Wikipedia data, which is a subset of the training corpus of BERT. Results are shown in Figure 1. From the left figure, AB-Net converges significantly faster than AB-Net FB, and AB-Net SC does not converge. When fine-tuning the whole model, more GPU memory is required because more gradient states need to be stored, therefore we have to halve the batchsize to fit the model into GPUs, which also slows down the training process. With the same batchsize, AB-Net saves 29% GPU memory and 26% wallclock training time compared with AB-Net FB. Moreover, we find that directly fine-tuning BERT is very unstable and sensitive to the learning rate, while only tuning the adapters can avoid this problem. In the right figure, the gains brought by BT are limited as adding over 1M monolingual data brings a performance drop. Also, BT requires to train another model and decode a large amount of monolingual data, which is time consuming. And our method is orthogonal with BT as shown by the Ro-En results in Table 3(a) of the main paper.

**To Reviewer 1:** We thank a lot for your detailed feedback!

**R1Q1:** The speedup of adapters and ablate more pre-training sizes. **R1A1:** Please refer to the common response for our analyses about the first point. For the second point, we have not explored the performance of our method on other pre-training models yet, but we do plan to consider small pre-training models such as DistilBert to explore more choices of adapters. We will also try to find appropriate case studies to provide intuitive illustrations.

**R1Q2:** Regarding the experimental settings. **R1A2:** We follow Zhu et.al. and use multi-bleu.perl to evaluate results for all models. The data in IWSLT14 tasks is lower cased in most previous works, so does Zhu et.al. according to their released code. Therefore we use uncased BERT for English and report the uncased BLEU scores in IWSLT14 tasks.

**To Reviewer 2:** We thank a lot for your insightful feedback! For your second question in "Additional feedback", We have tried different architecture variants of adapters and we find the results are similar.

**R2Q1:** For the suggestions on additional baselines. **R2A1:** Please refer to the common response for the comparisons between our method and additional baselines. For low-resource scenarios, as the multi-lingual BERT is trained on 104 languages, we believe our method can be directly applied to these languages. For other low-resource languages, if there exists monolingual data, then we can first pre-train BERT on it and then apply our method. If not, then it is naturally a very hard problem in MT which requires more ad-hoc techniques, and we will consider them as future works.

**To Reviewer 3:** We thank a lot for your positive comments about our work! Our method is model agnostic and can be applied to other pre-trained models with less memory and computation cost such as DistilBert.

**To Reviewer 4:** We thank a lot for your careful check of our manuscript! Regarding your questions in "Weaknesses":

**R4A1&R4A3:** Please refer to the common response for the additional comparisons with baselines. Our method efficiently brings more performance gains than back-translation, and is orthogonal with it. In addition, we can actually save GPU memory by only tuning the adapters rather than tuning the whole framework. **R4A2:** Please refer to R2A1 and Table 2 in the main paper where we have discussed about low-resource scenarios.

**R4A3:** We will revise our writing w.r.t your questions Q1-Q4 in "Additional feedback". For Q6&Q7, please refer to our common response and R4A2. For Q5, our method can additionally achieves a promotion of 0.2 BLEU scores on WMT14 En-De when initialized with a pre-trained Transformer decoder. We will update the results in the revision.

[Meta-Review · NeurIPS 2020]

This paper provides a novel approach to integrating a pre-trained model into a sequence-to-sequence model by decoupling the parameters of a light-weight adapter module from those of the pre-trained BERT models. The approach is task-agnostic and can be deployed in different problems and they show strong results in speedup and BLEU scores for non-autoregressive and auto-regressive machine translation. The reviewers all agreed that it is worth publishing. The author response was detailed and appropriate and there was some further reviewer discussion which led to a consensus accept. The paper is intuitive and simple and I think it will be an interesting addition to NeurIPS2020.